# A genome-wide association study of imaging-defined atherosclerosis

Anders Gummesson [1,2] ✉, Per Lundmark [3], Qiao Sen Chen [4], Elias Björnson[2], Koen F. Dekkers [3], Ulf Hammar[3], Martin Adiels[5], Yunzhang Wang [6], Therese Andersson[7], Göran Bergström [2,8], Carl-Johan Carlhäll[9,10,11], David Erlinge[12], Tomas Jernberg[6], Fredrik Landfors [7], Lars Lind[13], Maria Mannila [14], Olle Melander[15,16], Carlo Pirazzi[17], Johan Sundström[18,19], Carl Johan Östgren [9,11], Cecilia Gunnarsson [20], Marju Orho-Melander [15], Stefan Söderberg [7], Tove Fall [3] & Bruna Gigante [4,14]

Imaging-defined atherosclerosis represents an intermediate phenotype of atherosclerotic cardiovascular disease (ASCVD). Genome-wide association studies (GWAS) on directly measured coronary plaques using coronary computed tomography angiography (CCTA) are scarce. In the so far largest population-based cohort with CCTA data, we performed a GWAS on coronary plaque burden as determined by the segment involvement score (SIS) in 24,811 European individuals. We identified 20 significant independent genetic markers for SIS, three of which were found in loci not implicated in ASCVD before. Further GWAS on coronary artery calcification showed similar results to that of SIS, whereas a GWAS on ultrasound-assessed carotid plaques identified both shared and non-shared loci with SIS. In two-sample Mendelian randomization studies using SIS-associated markers in UK Biobank and CARDIoGRAMplusC4D, one extra coronary segment with atherosclerosis corresponded to 1.8-fold increased odds of myocardial infarction. This GWAS data can aid future studies of causal pathways in ASCVD.

The main pathological process underlying atherosclerotic cardiovascular disease (ASCVD) is lipid accumulation and low-grade inflammation in the vascular wall that leads to the formation of plaques. Atherosclerosis typically begins years before the first cardiovascular event and progresses from simple asymptomatic lesions to advanced plaques that may cause acute myocardial infarction (MI) or ischaemic stroke[1]. The identification of causal factors for atherosclerosis will increase our understanding of the mechanisms underlying ASCVD and enable the development of more effective and personalized treatments. To this end, imaging-defined atherosclerosis is a relevant outcome in genetic association studies, as it represents an intermediate phenotype whose progression precedes the debut of an overt clinical disease.

Genetic factors play a significant role in atherosclerosis, as shown by family history and heritability studies[2]. Much of our current knowledge about the genetics of ASCVD comes from genome-wide association studies (GWAS) that compare cases and controls for clinical endpoints such as MI and stroke[3–9]. The largest genetic association studies of atherosclerotic traits are meta-analyses that used the coronary artery calcium score (CACS), carotid intima-media thickness, or carotid plaques as main outcome variables[10–12]. Genetic studies on directly measured coronary artery plaques by contrast-enhanced coronary CT angiography (CCTA) are scarce and mainly involve hospital-based populations with symptomatic coronary artery disease (CAD), and thus do not include individuals who are healthy, who have subclinical CAD, or who died before the hospital[6].

The Swedish CArdioPulmonary bioImage Study (SCAPIS) is a prospective, population-based observational study with 30,154 individuals aged 50–64 years[13,14]. The comprehensive imaging protocol

includes coronary calcium scoring, CCTA, and carotid ultrasound examinations. In the present study, we used SCAPIS to perform a GWAS of imaging-defined atherosclerosis in the coronary and carotid arteries. The main aim was to elucidate the genetic basis of coronary plaque burden, defined as the segment involvement score (SIS) from CCTA, and to compare its genetics with that of CACS and carotid plaques (CarPlaq). The GWAS data was then used to investigate relationships between risk factors and atherosclerosis, and finally, to estimate the effect of genetically driven coronary plaque burden on MI.

## Results

### Study population

The clinical characteristics of the study population are shown in Supplementary Table S1. Of the 30,154 SCAPIS participants, there were 27,010 participants of European ancestry with available genotyping data after quality control (Supplementary Fig.S1). The mean age in this population was 57.6 (SD ± 4.4) years and 48.5% were males. The number of subjects with available data on both genotype and atherosclerosis traits were 24,811 for SIS, 26,000 for CACS, and 26,807 for CarPlaq. Of these, 10,589 (42.7%) had SIS > 0, 10,645 (40.9%) had CACS > 0 and 14,813 (55.3%) had CarPlaq > 0, respectively (Supplementary Table S1).

### GWAS results

Manhattan plots of the GWAS results for SIS, CACS and CarPlaq are shown in Fig.1A–C. No substantial bias in the association results were evident as observed values corresponded well with expected values in the QQ-plot (Fig.1D). Genomic inflation factor was 1.113 for SIS, 1.010 for CACS and 1.036 for CarPlaq. Significant independent single-nucleotide polymorphisms (SNPs) associated with SIS, CACS and CarPlaq are shown in Table 1, and further gene annotations of these SNPs are provided in Supplementary Table S2. For all lead SNPs associated with any outcome, associations with SIS, CACS and CarPlaq, together with data from three recent GWAS studies on CACS, CAD events and carotid intima-media thickness, are provided in Supplementary Data 1.

The SIS GWAS identified 20 SNPs, and of these, two SNPs had linkage disequilibrium (LD)-blocks overlapping with another independent SNP and hence merged into a common locus in the FUMA analysis (rs10455872, rs186696265: *LPA-PLG* locus, and rs2000660, rs9515203: *COL4A1-COL4A2* locus). Hence, the results for SIS corresponded to 18 different loci. The CACS GWAS identified 15 SNPs, of which two were found in the *LPA-PLG* locus, two in the *COL4A1-COL4A2* locus, two in the *APOE* locus, and the remaining in separate loci. The CarPlaq GWAS identified seven SNPs which were all in separate loci (Table 1).

There was a large overlap between the GWAS results for SIS and CACS, although fewer significant SNPs for CACS. When comparing beta coefficients for SIS and CACS, the results were similar across all significant SNPs (Fig.1E). Of the 18 loci for SIS, three were also associated with CarPlaq: *CDKN2B-AS1*, *LPA-PLG* and *APOE*. Of the seven loci for CarPlaq, *EDNRA*, *HDAC9*, *TMEM170A* and *KANK2* were not significant for either SIS or CACS. An overall comparison of beta coefficients between SIS and CarPlaq indicated both similarities and differences (Fig.1F), which were further investigated with colocalization analysis as described in the next section.

We also studied the SIS-associated SNPs in relation to localization and severity of coronary plaques. The associations were highly similar regardless of localization (i.e., right, circumflex, left, proximal or distal coronary arteries), while rs10455872 and rs186696265 in the *LPA-PLG* locus had higher beta-coefficients for severe stenosis ( > 50%) than for SIS (Supplementary Fig.S2).

### Shared and non-shared loci between coronary and carotid atherosclerosis

To identify putative causal genetic risk factors that are shared among atherosclerosis traits, single- and multivariant colocalization analyses were performed. Results from the single variant colocalization analysis between SIS and CarPlaq are shown in Fig.2A, indicating a high probability (PH4 > 0.7) of a shared genetic aetiology within a 250 kb region surrounding *LPA-PLG*, *TRIB1*, *CDKN2B*, *APOE* and *TMEM170A*-related variants. For the remaining SIS- and CarPlaq-associated variants, no clear colocalization was observed.

Variants showing a high probability of colocalization were then subject to fine-mapping and multivariant genetic colocalization analysis to narrow down the number of putative causal variants associated with each trait. Figure2B shows the results of the colocalization between SIS and CarPlaq for rs186696265 in the *LPA-PLG* locus. Within the 250 kb region surrounding rs186696265, we identified rs186696265 as the causal variant associated with SIS, but not with CarPlaq, while rs74617384, rs55730499, rs10455872 were the putative causal variants associated with both SIS and CarPlaq. Results from fine-mapping and multivariant colocalization analyses for the other loci are provided in Supplementary Figs.S3–S10.

### Effect mediators of significant SNPs

The role of key CVD risk factors as mediators for the genetic associations with SIS was assessed by adjusting for each SNP and risk factor separately (Fig.3). Associations were adjusted for the potential mediators body mass index (BMI), haemoglobin A1c (HbA1c), low-density lipoprotein cholesterol (LDL-C), lipoprotein(a) [Lp(a)], systolic blood pressure (SBP) and triglycerides (TG). Adjustment with Lp(a) attenuated the effect sizes for the two SNPs located in the *LPA-PLG* locus (rs10455872 and rs186696265). No other clear effect mediators were seen apart from a tendency towards lipid-mediation for 28601761 and rs7412.

### Genetic correlations

Genetic correlations across atherosclerosis variables and ASCVD-related traits using GWAS data generated within SCAPIS (Fig.4A) and data from external GWAS consortia (Fig.4B). The genetic correlations between atherosclerosis traits (SIS, CACS, CarPlaq and CAD events) were generally high, whereas for ASCVD risk factors such as BMI, SBP, HbA1c and lipids, the genetic correlations with atherosclerosis variables were relatively modest overall.

### Estimating the relationship between coronary plaque burden and MI using GWAS data

To study the effect of genetically driven coronary plaque burden on MI, we combined our GWAS results with data on genotype vs MI associations in UK Biobank and CARDIoGRAMplusC4D. In a Mendelian randomization analysis that included UK Biobank data, the odds ratio (OR) of having MI was 1.83 (95% CI: 1.59-2.10) per one extra coronary artery segment with atherosclerosis (Fig.5A). A similar result was obtained from CARDIoGRAMplusC4D (OR 1.76, 95% CI: 1.51-2.04) (Fig.5B). Sensitivity analysis in UK Biobank using the MR-Egger method, to account for potential pleiotropic effects, resulted in a higher estimate (OR 2.52, 95% CI: 1.89-3.37), hence it is possible that the IVW may slightly underestimate the true SIS-associated MI risk. For further details on sensitivity analyses, see Supplementary Table S3.

## Discussion

In the largest contemporary population study with CCTA data and several measures of atherosclerosis, we performed a GWAS on coronary plaque burden and identified 20 significant independent genetic markers for SIS, three of which were found in loci not implicated in ASCVD before. The GWAS results for CACS were highly similar to those

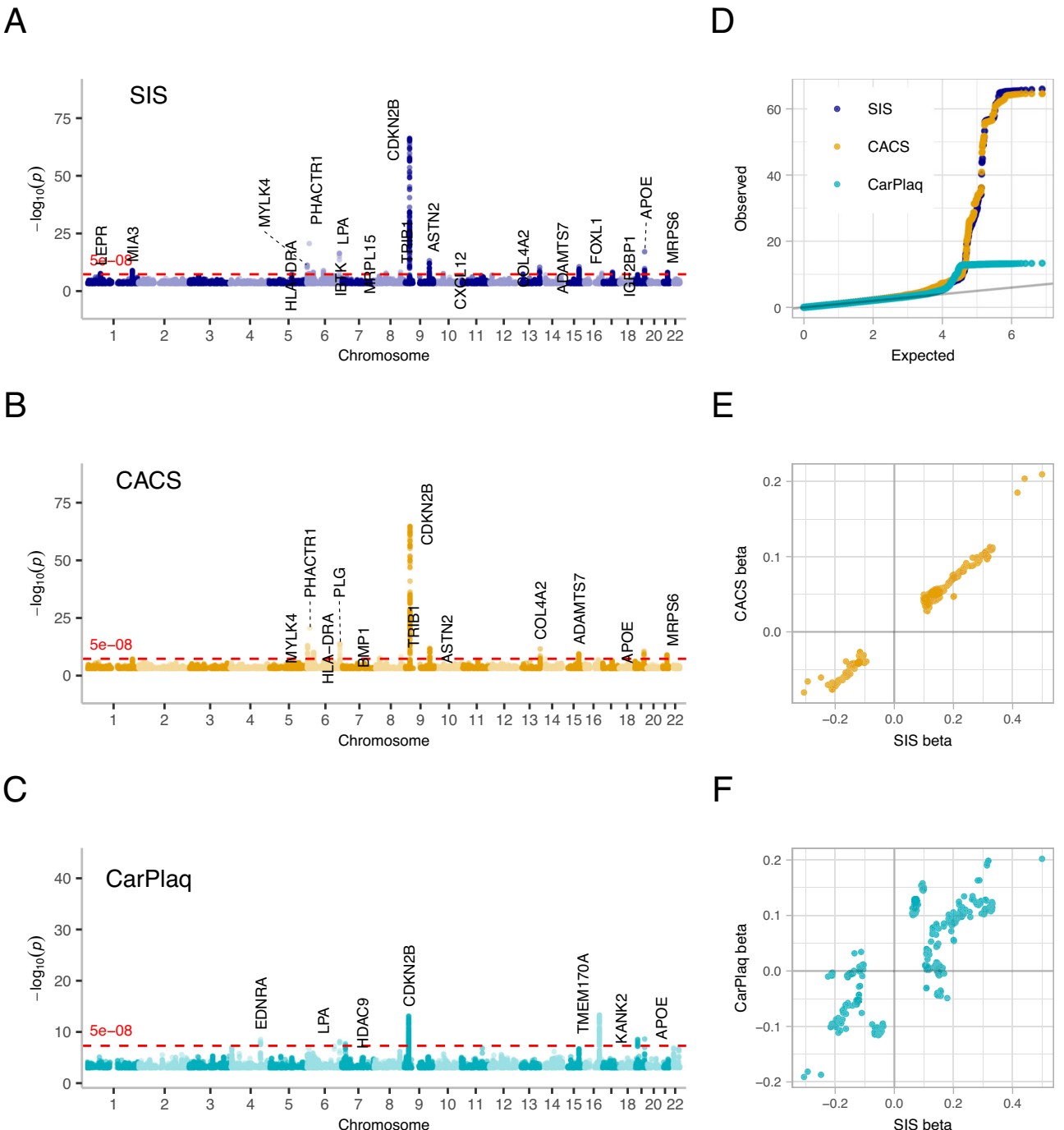

**Fig. 1 | GWAS results for atherosclerosis outcome variables. A–C** Manhattan plots of 2-sided unadjusted *p*-values for GWAS results for segment involvement score (SIS; blue, *n* = 24,811 participants), coronary artery calcium score (CACS; yellow, *n* = 26,000 participants), and carotid plaque (CarPlaq; turquoise, *n* = 26,807 participants), respectively. Plots depict unadjusted *p*-values below 0.001 from the GRAB/POLMM package for ordinal outcomes (SIS, CarPlaq) or Regenie software for continuous outcomes (CACS). Red dashed line depicts the genome-wide significant threshold of *p* = 5 × 10⁻⁸. **D** Quantile - quantile plot of observed vs. expected *p*-values, plotted against each other to highlight potential *p*-value inflation. Plotted are the 10,000 most extreme observed *p*-values for each outcome and a sample of 300,000 values per outcome. **E** Plots comparing beta coefficients between SIS and CACS for SNPs associated with at least one of the outcome variables. **F** Plots comparing beta coefficients between SIS and CarPlaq for SNPs associated with at least one of the outcome variables. Source data are provided as a Source Data file.

of SIS, whereas the comparison between SIS and carotid plaques revealed both shared and non-shared loci.

We identified 20, 15 and 7 independent SNPs for SIS, CACS and CarPlaq, respectively. Most of the SNPs identified are found in loci that have been implicated in ASCVD before, if all GWAS data including those on clinical events are considered. *CDKN2B-AS1* constituted the strongest signal in our data, which is consistent with previous GWAS of ASCVD including clinical events and imaging-defined atherosclerosis[3–7,9,11,12]. The lead variants in the *APOE*, *LPA-PLG*, *PHACTR1*, *TRIB1* and *COL4A1-COL4A2* loci have all been reported to associate with ASCVD[3–7,9,15]. The other variants are also found in loci that have been linked to ASCVD via nearby SNPs (within ±500 kb), except rs17127656, rs12190315 and rs2594811.

Of the novel SIS-associated SNPs, the strongest association was seen for rs12190315. While *MYLK4* is annotated as the closest protein-coding gene, rs12190315 is located in an intron of the long intergenic

**Table 1 | Significant independent SNPs from GWAS of segment involvement score (SIS), coronary artery calcium score (CACS), and carotid plaque (CarPlaq)**

| SNP | Chromosome:position:substitution (GRCh37) | Effect allele frequency | Nearest protein-coding gene | Beta (Standard error) | P-value |
|---|---|---|---|---|---|
| **SIS** | | | | | |
| rs17127656* | 1:65943471:C > T | 0.07 | LEPR | 0.202 (0.037) | 4.0e-08 |
| rs1909197 | 1:222819221:G > A | 0.81 | MIA3 | 0.144 (0.024) | 2.1e-09 |
| rs12190315* | 6:2626083:C > T | 0.22 | MYLK4 | −0.158 (0.023) | 9.7e-12 |
| rs9349379 | 6:12903957:A > G | 0.44 | PHACTR1 | 0.179 (0.019) | 4.3e-21 |
| rs9268807 | 6:32423915:C > G | 0.84 | HLA-DRA | 0.149 (0.026) | 8.6e-09 |
| rs1578654 | 6:82671826:T > C | 0.64 | IBTK | −0.119 (0.020) | 2.2e-09 |
| rs10455872 | 6:161010118:A > G | 0.07 | LPA | 0.313 (0.037) | 5.3e-17 |
| rs186696265 | 6:161111700:C > T | 0.01 | PLG | 0.500 (0.090) | 2.7e-08 |
| rs2594811* | 8:55213574:A > G | 0.44 | MRPL15 | 0.109 (0.020) | 3.0e-08 |
| rs28601761 | 8:126500031:C > G | 0.43 | TRIB1 | −0.110 (0.019) | 8.4e-09 |
| rs4977575 | 9:22124744:C > G | 0.45 | CDKN2B | 0.331 (0.019) | 1.1e-66 |
| rs3891689 | 9:119258583:T > C | 0.23 | ASTN2 | 0.168 (0.023) | 1.2e-13 |
| rs631414 | 10:44736284:C > T | 0.28 | CXCL12 | −0.116 (0.021) | 4.1e-08 |
| rs2000660 | 13:110788441:G > A | 0.09 | COL4A1 | −0.224 (0.037) | 1.4e-09 |
| rs9515203 | 13:111049623:T > C | 0.29 | COL4A2 | −0.135 (0.021) | 7.8e-11 |
| rs4243085 | 15:79057723:G > C | 0.73 | ADAMTS7 | −0.142 (0.022) | 4.7e-11 |
| rs423984 | 16:86713829:G > A | 0.36 | FOXL1 | 0.112 (0.020) | 2.3e-08 |
| rs2088140 | 17:47094599:T > C | 0.54 | IGF2BP1 | −0.108 (0.019) | 1.7e-08 |
| rs7412 | 19:45412079:C > T | 0.08 | APOE | −0.305 (0.036) | 1.5e-17 |
| rs9980618 | 21:35600505:C > T | 0.14 | MRPS6 | 0.157 (0.028) | 1.3e-08 |
| **CACS** | | | | | |
| rs12190315* | 6:2626083:C > T | 0.22 | MYLK4 | -0.058 (0.008) | 1.1e-13 |
| rs9349379 | 6:12903957:A > G | 0.44 | PHACTR1 | 0.061 (0.006) | 2.9e-21 |
| rs9268807 | 6:32423915:C > G | 0.84 | HLA-DRA | 0.058 (0.009) | 6.9e-11 |
| rs2315065 | 6:161108144:C > A | 0.08 | PLG | 0.095 (0.012) | 1.6e-14 |
| rs186696265 | 6:161111700:C > T | 0.01 | PLG | 0.209 (0.031) | 1.4e-11 |
| rs73225858 | 8:22049341:C > T | 0.06 | BMP1 | 0.076 (0.014) | 3.8e-08 |
| rs28601761 | 8:126500031:C > G | 0.43 | TRIB1 | −0.038 (0.007) | 6.4e-09 |
| rs10738606 | 9:22088090:A > T | 0.44 | CDKN2B | 0.112 (0.007) | 2.7e-65 |
| rs3891689 | 9:119258583:T > C | 0.23 | ASTN2 | 0.055 (0.008) | 1.4e-12 |
| rs11617955 | 13:110818102:T > A | 0.12 | COL4A1 | −0.060 (0.010) | 3.0e-09 |
| rs9515203 | 13:111049623:T > C | 0.29 | COL4A2 | −0.049 (0.007) | 3.1e-12 |
| rs4243085 | 15:79057723:G > C | 0.73 | ADAMTS7 | −0.046 (0.007) | 3.1e-10 |
| rs429358 | 19:45411941:T > C | 0.16 | APOE | 0.055 (0.009) | 2.0e-10 |
| rs7412 | 19:45412079:C > T | 0.08 | APOE | −0.081 (0.012) | 3.2e-11 |
| rs9305545 | 21:35595821:A > G | 0.17 | MRPS6 | 0.054 (0.009) | 1.3e-09 |
| **CarPlaq** | | | | | |
| rs78049276 | 4:148427503:A > C | 0.12 | EDNRA | 0.158 (0.027) | 2.8e-09 |
| rs74617384 | 6:160997118:A > T | 0.07 | LPA | 0.199 (0.034) | 6.1e-09 |
| rs2074633 | 7:19035920:T > C | 0.23 | HDAC9 | 0.113 (0.020) | 1.8e-08 |
| rs1537370 | 9:22084310:C > T | 0.43 | CDKN2B | 0.128 (0.017) | 7.1e-14 |
| rs7198873 | 16:75474754:G > A | 0.59 | TMEM170A | 0.129 (0.017) | 4.3e-14 |
| rs4423524 | 19:11275394:C > A | 0.43 | KANK2 | −0.103 (0.017) | 2.5e-09 |
| rs7412 | 19:45412079:C > T | 0.08 | APOE | −0.191 (0.032) | 2.1e-09 |

P-values represent unadjusted 2-sided p-values from GRAB/POLMM or Regenie software with a genome-wide significance threshold of p < 5e-8. The symbol * denotes SNPs in loci not previously implicated in ASCVD.

non-protein coding RNA 1600 (LINC01600) and is associated with LINC01600 expression in arterial tissue in the GTEx database. This could be of interest to pursue further given the growing body of evidence linking various LINC-genes to the pathogenesis of atherosclerosis[16]. The SNP rs17127656 is an intron variant in the LEPR gene, and rs2594811 is an intergenic variant between the MRPL15 and RNU105C genes. To the best of our knowledge, these loci have not emerged as significant ASCVD risk loci before. In a look-up of these SNPs using CACS GWAS data by Kavousi et al, only rs12190315 had a low (albeit not genome-wide significant) p-value of $6.3 \times 10^{-6}$

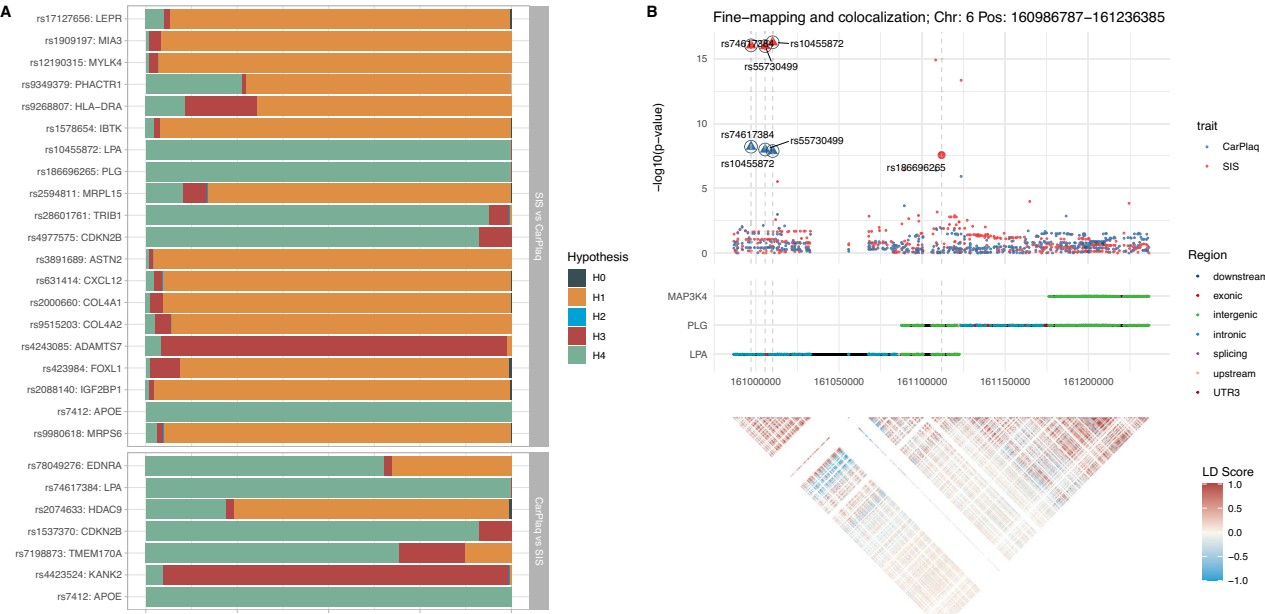

**Fig. 2 | Colocalization analyses. A** Single variant colocalization for significant independent SNPs from segment involvement score (SIS) and carotid plaque (CarPlaq) GWAS. This analysis is based on the log approximate Bayes factor, calculating five posterior probabilities for five hypotheses: H0 (blue): none of the variants are associated with any of the traits; H1 (orange) and H2 (turquoise): one variant associates with one or the other trait; H3 (red): two different variants associate with each of the two traits; H4 (green): one variant associates with both traits. **B** Representative scatter plot of the multivariant colocalization between SIS and CarPlaq using the Sum of Single Effects (SuSiE). Here, genetic variants within a window of 250 kb in the locus near rs186696265 (*PLG* locus) were fine-mapped for SIS and CarPlaq. In the scatter plot, one set of genetically colocalized SNPs between CarPlaq (blue) and SIS (red) are identified as triangles marked by black circles. Enlarged points represent fine-mapped SNPs for each trait. Below the scatter plot, the middle panel displays a line plot showing the genes and region associated with each SNP. At the bottom, the triangular heatmap shows the linkage disequilibrium (LD) structure for the region surrounding rs186696265. Source data are provided as a Source Data file.

(Supplementary Data 1)[11]. Hence, additional studies in independent cohorts with CCTA data would be needed to validate our findings.

The GWAS results for SIS and CACS were highly similar, although more markers reached genome-wide significance for SIS than for CACS, likely due to better classification. While these are both measurements of coronary atherosclerosis, SIS derived from CCTA is a direct measurement of plaque burden whereas CACS is an indirect proxy. Indeed, we have recently shown that 5.5% of SCAPIS-participants with CACS = 0 had coronary atherosclerosis detected by CCTA[14]. However, cohorts with CCTA data are scarce, which hampers the possibility of increasing statistical power by combining cohorts in GWAS meta-analyses. Our finding that the genetics of CACS reflects that of SIS is therefore reassuring for future GWAS meta-analyses that use CACS to estimate coronary atherosclerosis. Although we did not detect any major difference in the GWAS results between the two measurements, it is noteworthy that the only SNP that reached genome-wide significance for CACS but not SIS was an intron variant in *BMP1*, a gene previously implicated in vascular calcification[17].

Coronary and carotid atherosclerosis share many similarities with regards to risk factors and pathophysiology[18]. The SCAPIS cohort offers the opportunity to directly compare the genetics of atherosclerosis in the two vascular beds, thereby removing various bias that may arise when comparing data from separate cohorts. We found that the three well-known CVD risk loci (*CDKN2B-AS1*, *LPA-PLG* and *APOE*) were associated with both coronary and carotid atherosclerosis, but there were also considerable differences in the genetic background for atherosclerosis between the two vascular beds: The majority of SIS-associated SNPs did not appear to colocalize with CarPlaq, and the overall genetic correlation between SIS and CarPlaq was relatively modest ($r = 0.42$).

Several of the identified loci have been implicated in ASCVD risk factors such as circulating lipids (*LPA-PLG*, *APOE*, *TRIB1*, *KANK2*),

diabetes (*CDKN2B-AS1*, *IGF2BP1*) and obesity (*LEPR*, *IGF2BP1*). This prompted us to investigate these risk factors as potential mediators of genetic associations. The only clear mediating effects of established risk factors in our data were circulating Lp(a) levels for the *LPA-PLG* locus. In addition, the overall genetic correlations between atherosclerosis variables and risk factors were relatively modest. Taken together, these findings support previous data indicating that genetics can capture risk information that is largely independent of traditional risk factors[19].

The development of coronary artery plaques is known to be the main pathophysiological process towards MI. SIS is a direct measure of coronary plaques, and its role as predictor of coronary events needs to be further validated. Here, we investigated the relationship between coronary plaque burden and risk of MI, which to our knowledge has not been quantified previously using genetic data. By comparing the genetic effect on SIS and MI in a Mendelian randomization framework, we found that a genetic propensity for one extra coronary artery segment with atherosclerosis corresponded to a 1.8-fold increased risk of MI. To put this estimate into context, a previous meta-analysis of epidemiological studies reported a pooled hazard ratio of 1.25 for major cardiovascular events per segment increase in SIS[20]. While the differences in study designs do not allow for a direct comparison of results, our estimate from genetically driven plaque burden appear higher than expected based on epidemiological data. Of note, our sensitivity analyses with MR-Egger indicate that the OR from the main analysis may be a conservative estimate rather than being upward biased.

Another potential application for these GWAS data, outside the scope of the present study, is to use GWAS summary statistics to develop polygenic risk scores (PRS). Most current PRS for CAD are constructed from hard events, but for applications that aim to identify

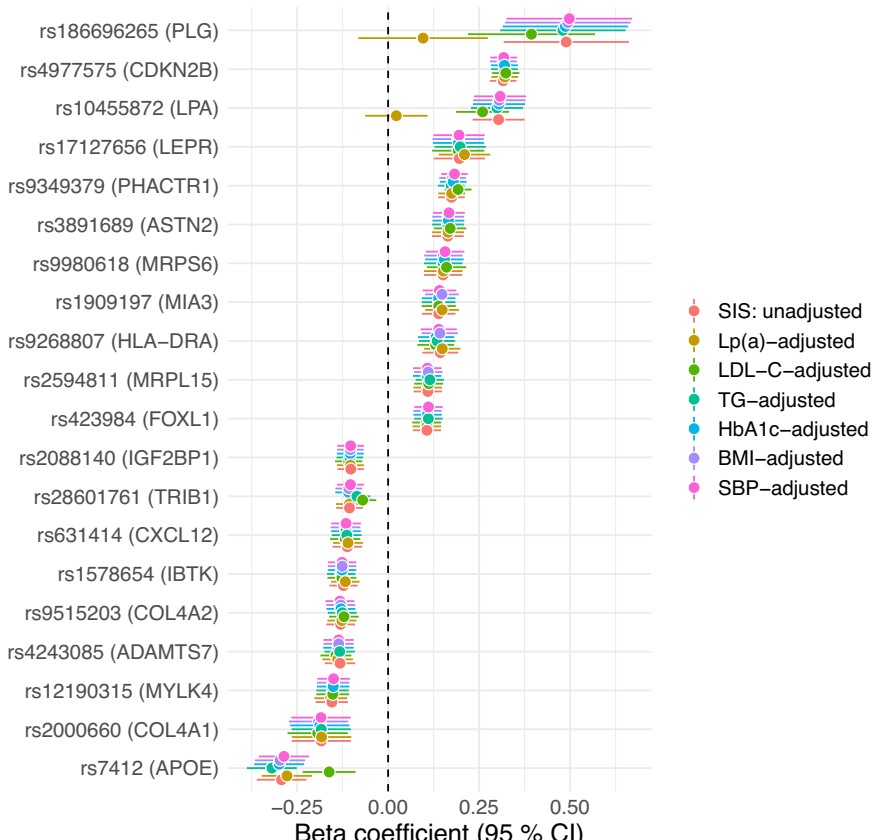

**Fig. 3 | Analysis of potential effect mediators for each of the 20 SNPs associated with segment involvement score (SIS).** Associations (beta-coefficients between SNPs and SIS, derived from *n* = 24,811 participants) were adjusted for the potential mediators lipoprotein(a) [Lp(a)], low density lipoprotein cholesterol (LDL-C), triglyceride (TG), haemoglobin A1c (HbA1c), body mass index (BMI), and systolic blood pressure (SBP). Dots represent beta-coefficients and error bars represent 95% confidence intervals. Source data are provided as a Source Data file.

individuals with subclinical atherosclerosis, a more plaque-specific PRS could be evaluated.

Our study is based on a predominantly Swedish population in the age span of 50–64 years, which may limit generalizability. More studies in other populations are therefore needed. Although detectable coronary plaques were common (42.7%), most were of mild to moderate severity and relatively few had obstructive CAD (5.7%). Nevertheless, also mild to moderate atherosclerosis is of relevance as it confers a considerably increased risk of CVD events[21]. Most importantly, genetic association data of mild atherosclerosis may help to elucidate mechanisms involved in the early phases of atherosclerosis development.

SCAPIS is by far the largest population-based cohort with CCTA-data, which is a major strength of this study but also limits the possibilities for external validation of the novel genetic associations with SIS. Another limitation was that carotid intima-media thickness was not available as an outcome variable. However, previous studies have shown that the presence of carotid plaques is a more powerful predictor of ASCVD events than intima-media thickness[22].

In conclusion, this GWAS of imaging-defined atherosclerosis identified 20 significant independent SNPs associated with coronary plaque burden as measured using CCTA. The genetic basis of SIS was highly similar to that of CACS, while considerable differences were seen between coronary and carotid atherosclerosis. When estimated from genetic data, one extra coronary artery segment with atherosclerosis corresponded to a 1.8-fold increased risk of MI. This GWAS data can be used as a resource to inform Mendelian randomization studies of causal pathways between risk factors, atherosclerotic plaque burden and clinical events.

## Methods

### Ethics
This research complies with all relevant ethical regulations. SCAPIS was approved by the Ethical Review Board at Umeå University, Sweden (Dnr 2010-228-31M, Dnr 2017-183-31M, and Dnr 2021-00371). The genetics analysis in SCAPIS was approved by the Swedish Ethical Review Authority (Dnr 2020-04923). All study participants provided informed written consent before enrolment in the study.

### Study population
SCAPIS (https://scapis.org) was initiated as a major joint national effort in Sweden to study mortality and morbidity from cardiovascular disease, chronic obstructive pulmonary disease and related metabolic disorders and risk factors[13]. Between 2013 and 2018, 30,154 subjects aged 50–64 were randomly recruited from the national population register in six areas adjacent to six Swedish university hospitals. No exclusion criteria were applied except the inability to understand written and spoken Swedish for informed consent. All participants followed a core programme consisting of at least two visits to the test centre, including imaging of the coronary and carotid arteries, comprehensive questionnaires, biochemistry, anthropometry, and blood pressure measurements.

### Coronary atherosclerosis
CCTA was performed using a dedicated dual-source scanner with a Stellar Detector (Somatom Definition Flash, Siemens Medical Solutions)[14]. Non-contrast, ECG-gated images for calcium scoring were obtained from all subjects, followed by contrast-enhanced (iohexol) images from subjects without contraindications to CCTA.

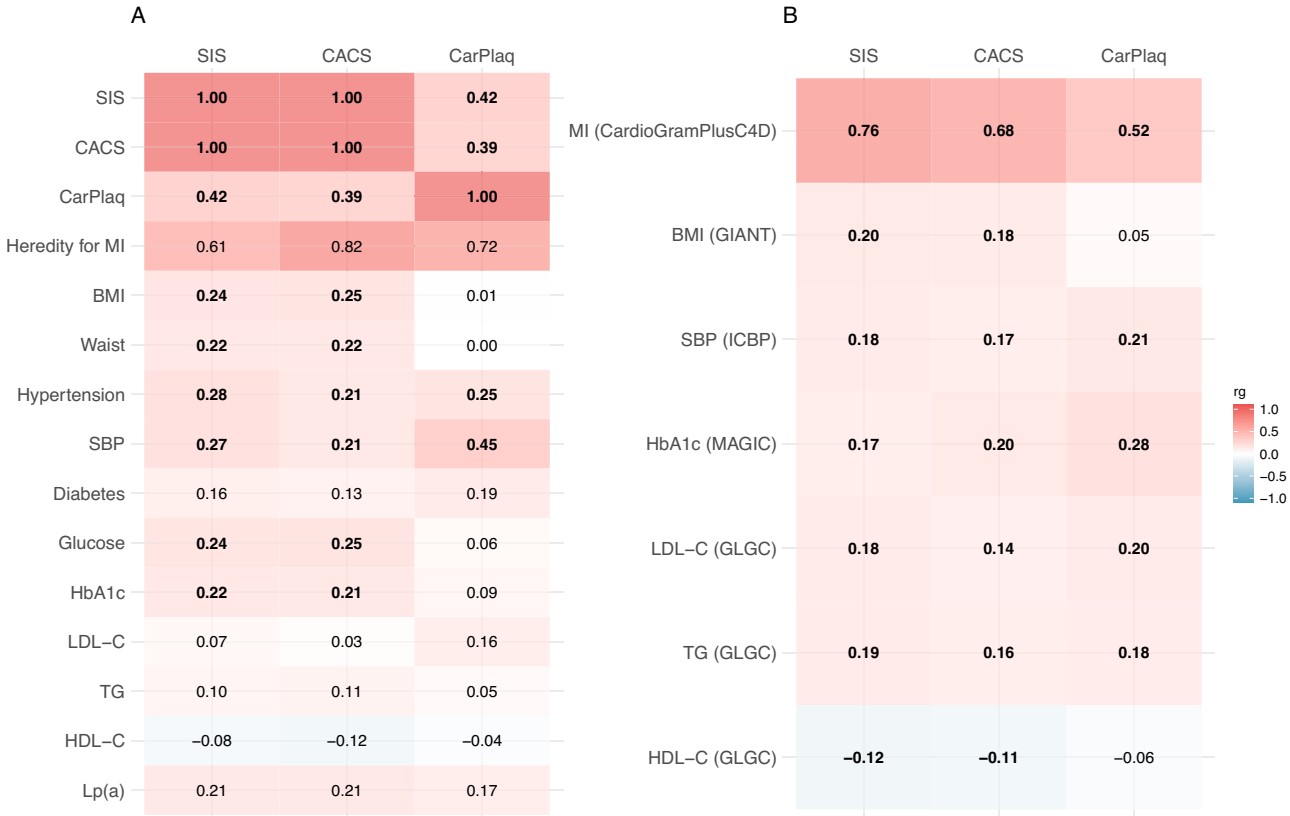

**Fig. 4 | Genome-wide genetic correlation ($r_g$) between atherosclerosis variables and other atherosclerosis-related phenotypes. A** Genetic correlations with phenotypes within SCAPIS. **B** Genetic correlations using GWAS data from external consortia. The $r_g$ value was estimated by linkage disequilibrium score regression. Numbers in bold refer to genetic correlations with two-sided $P$-value < 0.05, without adjustments for multiple comparisons. Source data are provided as a Source

Data file. SIS segment involvement score, CACS coronary artery calcium score, CarPlaq carotid plaque, MI myocardial infarction, BMI body mass index, SBP systolic blood pressure, HbA1c haemoglobin A1c, LDL-C low density lipoprotein cholesterol, TG triglyceride, HDL-C high density lipoprotein cholesterol, Lp(a) lipoprotein(a).

Calcifications were segmented on non-contrast images using the syngo.via calcium scoring software (Siemens, Erlangen, Germany) and summed for the whole artery tree to a coronary artery calcium score (CACS) according to Agatston criteria[23].

Information on atherosclerosis was reported from CCTA scanning using the 18 coronary segment model from the Society of Cardiovascular Computed Tomography[24]. To streamline reading and increase quality of the most important findings, readers focused on the 11 clinically most relevant segments (segments 1 through 3, 5 through 7, 9, 11 through 13, and 17), which were compulsory to report. For individuals with valid readings in at least four proximal segments, SIS was calculated as the total number of clinically relevant segments with atherosclerotic plaque.

### Carotid atherosclerosis
Carotid atherosclerosis (plaques ≥ 1.5 mm thickness) was categorized as absent, unilateral, or bilateral from 2-dimensional grayscale ultrasound images obtained with an Acuson S2000 ultrasound scanner and a 9L4 linear transducer (both from Siemens) using a standardized protocol and the Mannheim consensus[13,25].

### Cardiovascular risk factors
Data on age, sex, smoking habits, medical history, medication, and heredity for MI (defined as parent with MI before 60 years-of-age) were collected using standardized questionnaires. BMI was calculated from height and body weight measured with subjects dressed in light indoor clothing without shoes. Waist circumference was measured to the nearest cm midway between the palpated iliac crest and the palpated lowest rib margin. SBP and diastolic blood pressure (DBP) were

measured in the supine position twice in each arm with an automatic device (Omron M10-IT. Omron Health care Co. Kyoto. Japan) using the mean of SBP and DBP, respectively, from the arm with the higher mean SBP.

A venous blood sample was collected from participants after overnight fasting and used for immediate analysis of glucose, HbA1c, LDL-C (calculated using the Friedwalds formula), high-density lipoprotein cholesterol (HDL-C), and TG. Lp(a) was measured in biobanked plasma samples stored in -80 °C, using the Alinity C assay (Abbott Laboratories, IL, USA). Diabetes diagnosis was defined as fasting plasma glucose ≥ 7.0 mmol/L or HbA1c ≥ 48 mmol/mol ( > 6.5%) or previously known diabetes. Hypertension was defined as SBP > 140 or DBP > 90 mmHg, or medication for hypertension.

### Genotyping
Out of the 30,154 individuals in the SCAPIS cohort, 29,433 blood samples from individuals who had provided consent for genetic analyses were submitted for DNA extraction at Karolinska Institutet biobank (ki.se/en/research/ki-biobank). DNA was extracted from whole blood in a randomized order.

DNA samples were genotyped in 10 batches using a customized version of the Illumina GSA-MDv3 genotyping array (www.illumina.com) at the SNP&SEQ Technology Platform in Uppsala (www.genotyping.se), a part of the National Genomics Infrastructure Sweden and Science for Life Laboratory. Genotypes were called with the Illumina GenomeStudio 2.0.3 software, where genotype clusters were defined in the first batch ($n$ = 3033) and then applied to subsequent genotyping batches for consistency in cluster limits. On delivery, genotyping batches were monitored for genotype/phenotype sex

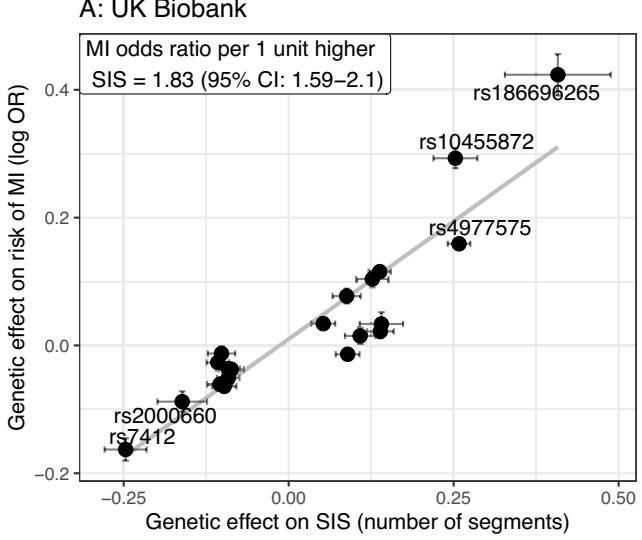

**Fig. 5 | Relationship between coronary plaque burden and myocardial infarction (MI). A, B** shows the relationship between genetically induced change in segment involvement score (SIS; quantified as a continuous variable in $n = 24{,}811$ SCAPIS participants) and MI evaluated in UK Biobank ($n = 487{,}202$) and CARDIoGRAMplusC4D ($n = 60{,}811$ cases and 123,504 controls), respectively. Dots represent beta-coefficients (beta-SIS on x-axis and log odds ratio for MI on y-axis), and error bars represent standard errors. Source data are provided as a Source Data file.

discordance, marker and sample missingness, heterozygosity rates, and batch effects related to DNA plate. The total autosomal genotyping rate was 0.998 and in total the dataset contained 29,431 genotyped samples and 726,358 markers before filtering.

**Imputation**
An input dataset for imputation was created by filtering genotypes for call rates in markers and individuals in two stages, first at 90%, then at 98%, Hardy-Weinberg Equilibrium (HWE) test at $1 \times 10^{-8}$ in Swedish origin samples and a minor allele frequency (MAF) of $> 0.1\%$. The data was then processed with Will Rayner's imputation preparation script (HRC-1000Gcheck-bim-v4.3.0, www.well.ox.ac.uk/~wrayner/tools) to check reference panel overlap, strand, alleles, positions, reference / alternative allele assignments and frequency differences. Palindromic SNPs were removed if minor allele frequency was $> 0.4$ to reduce the risk of markers with allele switches. SNPs with other alleles than the corresponding SNP in the reference panel were also removed, together with SNPs with more than a 0.2 allele frequency difference between data and reference panel. These steps resulted in a dataset of 29,429 samples and 538,960 markers. This data was then checked for sex genotype/phenotype discordance, where 4 samples were discarded, leaving 29,425 samples for imputation.

The genotype data was imputed to the HRC r1.1 reference panel (www.haplotypereference-consortium.org) at the Sanger imputation service (imputation.sanger.ac.uk)[26] with the pipeline "Pre-phasing and imputation with EAGLE2 + PBWT" (Eagle v2.4, PBWT v3.1). The HRC panel contains 40.4 M variants, and of these, 7,906,639 remained after filtering for imputation information score $\geq 0.7$ and MAF $\geq 0.01$.

**Filtering of imputed genotype data**
The filtering to create the final GWAS genotype data of 27,010 samples from the imputed dataset is shown as flowchart in Supplementary Fig.S1 and included the following steps:

Samples with sex aneuploidy ($n = 11$): Markers were pruned for linkage disequilibrium with plink 1.9[27] as "plink –indep-pairwise 50 5 0.2", and sex cheques performed with "plink –check-sex ycount". Samples were labelled as genotypic females if F < 0.75 and Y count < 500 and males if F > 0.75 and Y-count > 500. If samples had female F-values and male Y-counts they were called as XXY, and female Y-count with male F-values were called as X0 and removed from the data set.

Non-European ancestry ($n = 2048$): Subjects of non-European ancestry were excluded to avoid confounding by population stratification. Common variants between the study samples and the 1000 genomes[28] were extracted and the study samples merged with the 1000 genomes data set. Principal component analysis (PCA) was run on LD-pruned data with minor allele frequency > 5% filtered for long-range high LD regions[29,30]. Principal components (PCs) were calculated in unrelated samples and then all samples projected onto these PCs. Boundaries were set in PCs 1, 3 and 4 to filter out samples with non-European origin according to reported parental birth country and comparisons to 1000 genomes reference samples.

Heterozygosity within 3 standard deviations (SD) of the mean in the European samples ($n = 239$): The raw heterozygosity was calculated on LD-pruned data as the proportion of non-missing genotypes that are heterozygous, based on output from "plink –het" (h = (Nnm-Nhom)/Nnm). An adjusted heterozygosity was then calculated by adjusting the raw heterozygosity for geographic origin by a quadratic model with interaction terms using the six first PCs in the 1000 genomes / SCAPIS PCA. The adjusted heterozygosity was then used for filtering purposes. Samples were removed if an adjusted heterozygosity fell outside 3 SD from the mean.

Sample genotype call rate < 99% ($n = 117$): A strict sample call rate threshold was implemented due to the excellent performance of the samples in genotyping.

**GWAS analyses**
In the primary analysis, coronary atherosclerosis was defined by SIS as an ordinal variable (0–11 affected segments). CACS was rank-based inverse normal transformed due to its non-normal distribution. CarPlaq was defined as an ordinal variable (0=absent, 1=unilateral, 2=bilateral).

The GWAS analyses employed methods that correct for population structure and cryptic relatedness. Associations with ordinal outcome variables (SIS and CarPlaq) were tested using mixed models in the POLMM method implemented in the GRAB R-package v0.0.3.6 (wenjianbi.github.io/grab.github.io)[31]. A dataset for constructing the dense genetic relationship matrix, used to adjust for the relatedness among samples, was created by filtering the directly genotyped dataset for sample and marker missingness at 0.01, HWE at $1 \times 10^{-15}$, MAF at 0.01 and then LD pruning the markers with plink2 --indep-pairwise in

500 kb windows with an r2 threshold of 0.2. The POLMM null model was then fitted with options LOCO = FALSE, tolTau = 0.2 and tolBeta = 0.01. Single marker tests were used for the association analysis.

Continuous or rank-based inverse normal transformed variables and binary variables were analysed using the Regenie software v3.2.5 (rgcgithub.github.io/regenie)[32]. It uses a combination of whole genome regression and linear or logistic regression. A data set for the first step of the Regenie analysis adjusting for sample relatedness was created as above but without LD pruning.

The genetic exposure was modelled as linear, using dosages from the imputation procedure. Covariates included sex, age, study site, and the 10 components with the highest eigenvalues from principal component analysis of the genotype data. Age was modelled with restricted cubic splines (using four knots at the 5th, 35th, 65th and 95th percentile) to relax the assumption of linearity and thereby potentially improve power[33]. Potential systematic bias in the association results were assessed using quantile-quantile (Q-Q) plots and the genomic inflation (lambda) factor.

### Definition of significant independent SNP and locus
Significant independent SNPs were defined using the stepwise selection procedure implemented in GCTA-COJO[34], using LD data from the samples together with a collinearity cutoff of 0.9 and a p-value cutoff for inclusion of $5 \times 10^{-8}$.

Genomic risk loci were then defined by supplying the GCTA-COJO lead SNPs and GWAS results to FUMA GWAS (fuma.ctglab.nl) and utilizing their algorithm to define detected loci[35]. Variants in LD $r^2 > 0.6$ with one of the lead SNPs and $p < 0.05$ were collected in an LD block. Blocks in LD with a lead SNP within 250 kb were then merged into one locus. Annotation of loci in the main tables and figures were based on the protein-coding gene with the shortest distance to the lead SNP. Gene annotations were also obtained from Open Targets (https://www.opentargets.org)[36], Genotype-Tissue Expression (GTEx, https://www.gtexportal.org/home)[37], and Stockholm–Tartu Atherosclerosis Reverse Network Engineering Task (STARNET, http://starnet.mssm.edu)[38] (Supplementary Table S2).

### Colocalization and fine-mapping
To colocalize genetic variants associated with both coronary and carotid atherosclerosis, we performed single- and multivariant colocalization analysis in sequential order. For both single variant and multivariant genetic colocalization analyses, we set a non-informative prior for estimating the posterior probability of each hypothesis. In the single variant colocalization analysis, a region of 250Kb surrounding each independent significant SNP for SIS and CarPlaq was tested for colocalization between the traits by estimating the Bayes factor of colocalization using the R package "coloc"[39]. This package estimates five *a posteriori* probabilities for five hypotheses: H0: none of the variants are associated with any of the traits; H1 and H2: one variant associates with one or the other trait; H3: two different variants associate with each of the two traits; H4: one variant associates with both traits. We used H4 > 0.7 as a threshold to define a putative colocalization.

In the multivariant colocalization analysis, variants with a H4 > 0.7 were fine-mapped for each of the traits using the Sum of Single Effects (SuSiE). Briefly, the 250 kb window surrounding SNPs that colocalized across the traits were fine-mapped to narrow down the number of putative causal variants associated with each pair of traits. The multi-variant colocalization analysis was performed by estimating Bayes factors surrounding the fine-mapped SNPs from each atherosclerosis measure using the SuSie function (coloc.susie)[40].

### Assessment of potential effect mediators
To assess whether traditional risk factors could account for the SNPs association with SIS, additional GWAS analyses were performed with adjustment for Lp(a), LDL-C, triglycerides, SBP, BMI and HbA1c respectively. These traditional risk factors were included as covariates in the GWAS, and in the case of LDL-C, additional adjustment for lipid lowering medication was performed.

### Genetic correlations
We estimated pairwise genetic correlation ($r_g$) across SIS, CACS, CarPlaq, and other ASCVD-related variables. The $r_g$ value was estimated by LD Score (LDSC) regression[41], which measures the heritability correlation between two traits, aiming to isolate the genetic component from potential environmental influences. GWAS data from SCAPIS, as well as external consortia, including GLGC[42], GIANT + UK Biobank[43], MAGIC[44], ICBP + UK biobank[45], and CardiogramPlusC4D[5], were used for these analyses. The LD score reference was estimated based on the 1000 genomes European reference population.

### Estimating the relationship between coronary plaque burden and MI
Mendelian randomization was performed to estimate the causal relationship between SIS and MI. To obtain SNP vs outcome data, the association with MI was quantified in the UK Biobank cohort for the SNPs independently associated with SIS in SCAPIS. Among 487,202 subjects in UK Biobank with available genetic data, 27,783 cases had a fatal or non-fatal MI outcome using the definition provided by the UK Biobank Outcome Adjudication Group (https://biobank.ndph.ox.ac.uk/showcase/showcase/docs/alg_outcome_main.pdf). Non-fatal MI was based on medical history and 12 years of follow-up (6577 prevalent and 17,356 incident events). Fatal MI was based on 12 years of follow-up (3850 incident events). Associations between SNPs and MI were calculated using logistic regression models with adjustments for age, sex, and genetic principal components 1–5. UK Biobank has been approved by the North-West Multi-centre Research Ethics Committee as a Research Tissue Bank.

Data on associations with MI was also obtained from the CARDIoGRAMplusC4D 1000 Genomes-based GWAS meta-analysis (60,801 CAD cases and 123,504 controls), using a sub-group analysis in cases with a reported history of MI (~70% of total number of cases)[5].

Causal odds ratios between SIS and MI were quantified with the R-package "MendelianRandomization" using the inverse-variance weighted (IVW) method, and further sensitivity analyses using MR-Egger. Since there is already strong prior probability that coronary plaques are causally related to MI, the purpose of the Mendelian randomization analysis was primarily to quantify this relationship. Hence, to obtain an estimate expressed as the risk of MI per one extra segment with atherosclerosis, SIS was coded as a continuous variable in this analysis. The slope of the regression line between change in SIS and change in MI risk can be interpreted as the causal risk increase per one additional coronary segment with plaque.

### Reporting summary
Further information on research design is available in the Nature Portfolio Reporting Summary linked to this article.

## Data availability
The GWAS summary data for imaging-defined atherosclerosis have been deposited in the GWAS Catalogue database under accession codes GCST90503074, GCST90503075 and GCST90503076 (https://www.ebi.ac.uk/gwas/studies/GCST90503074, https://www.ebi.ac.uk/gwas/studies/GCST90503075, https://www.ebi.ac.uk/gwas/studies/GCST90503076). SCAPIS data are not publicly available due to privacy and ethical restrictions. Access to pseudonymized SCAPIS phenotype and genotype data requires ethical approval from the Swedish Ethical Review Board and approval from the SCAPIS Data access board (https://www.scapis.org/data-access/). All phenotype and genotype data for the UK Biobank are available to researchers with approved data

access from the UK Biobank (https://www.ukbiobank.ac.uk/enable-your-research/register). GLGC GWAS data are available using the following links for LDL-C https://csg.sph.umich.edu/willer/public/glgc-lipids2021/results/ancestry_specific/LDL_INV_EUR_HRC_1KGP3_others_ALL.meta.singlevar.results.gz, HDL-C https://csg.sph.umich.edu/willer/public/glgc-lipids2021/results/ancestry_specific/HDL_INV_EUR_HRC_1KGP3_others_ALL.meta.singlevar.results.gz, and TG https://csg.sph.umich.edu/willer/public/glgc-lipids2021/results/ancestry_specific/logTG_INV_EUR_HRC_1KGP3_others_ALL.meta.singlevar.results.gz. GIANT + UK Biobank GWAS data for BMI is available at https://portals.broadinstitute.org/collaboration/giant/images/c/c8/Meta-analysis_Locke_et_al%2BUKBiobank_2018_UPDATED.txt.gz. MAGIC GWAS data for HbA1c is available at http://magicinvestigators.org/downloads/MAGIC1000G_HbA1c_EUR.tsv.gz. CARDIoGRAMplusC4D GWAS data for MI is available at https://www.cardiogramplusc4d.org/media/cardiogramplusc4d-consortium/data-downloads/mi.additive.Oct2015.pub.zip. Source data are provided with this paper.

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

## Acknowledgements

The main funding body of the Swedish CArdioPulmonary bioImage Study (SCAPIS) is the Swedish Heart-Lung Foundation. The study is also funded by the Knut and Alice Wallenberg Foundation, the Swedish Research Council, VINNOVA (Sweden's Innovation agency), the University of Gothenburg and Sahlgrenska University Hospital; Karolinska Institutet and Region Stockholm, Linköping University and University Hospital, Lund University and Skåne University Hospital, Umeå University and University Hospital, Uppsala University and University Hospital. We would like to acknowledge the help of Biobank Sweden and the local biobank facilities for their services in handling of biological samples and biobanking. Genotyping was performed by the SNP&SEQ Technology Platform in Uppsala. The facility is part of NGI Sweden and Science for Life Laboratory. The SNP&SEQ Platform is also supported by the Swedish Research Council and the Knut and Alice Wallenberg Foundation. The computations and data handling were made possible by resources from project sens2019512 provided by the Swedish National Infrastructure for Computing (SNIC) at Uppsala Multidisciplinary Centre for Advanced Computational Science (UPPMAX), partially funded by the Swedish Research Council through grant agreement no. 2018-05973. We would like to acknowledge the Karolinska Institute Biobank for their services regarding DNA extraction. Research using the UK Biobank Resource has been conducted under Application Number 82018. GWAS data on MI have been contributed by CARDIoGRAMplusC4D investigators and was downloaded from www.CARDIOGRAMPLUSC4D.org. Data on glycaemic traits have been contributed by MAGIC investigators and were downloaded from www.magicinvestigators.org. We also acknowledge the investigators from the ICBP, GIANT, GLGC, STARNET, and GTEx consortia. Links to external GWAS summary statistics are provided in the Data Availability Statement. This study received financial support from the Swedish Heart-Lung Foundation (2023-0439 to B.G.; 2024-1135 and 2024-1137 to A.G.), the Swedish Research Council (2023-02177 to B.G.), and the Swedish state under the agreement between the Swedish Government and the county councils, the ALF-agreement (ALFGBG-991828 to A.G.).

## Author contributions

Conceptualization: A.G., P.L., E.B., Q.S.C., T.F., B.G.; data collection: G.B., C.J.C., D.E., T.J., M.M., L.L., O.M., J.S., C.J.Ö., C.P., C.G., M.O.M., T.F., A.G., B.G., S.S.; data analysis and visualization: P.L., M.A., E.B., Q.S.C., K.F.D., U.H., Y.W.; manuscript draft: A.G., P.L., E.B., Q.S.C., T.F., B.G., S.S., T.A., F.L.; critical revision of analysis plan and manuscript: all authors.

## Funding

## Competing interests

The authors declare the following competing interests: S.S. reports speakers' honoraria and participation in the scientific board for the Exposure study (Janssen). The remaining authors declare no competing interests.

## Additional information

[1]Region Västra Götaland, Sahlgrenska University Hospital, Department of Clinical Genetics and Genomics, Gothenburg, Sweden. [2]Department of Molecular and Clinical Medicine, Institute of Medicine, Sahlgrenska Academy, University of Gothenburg, Gothenburg, Sweden. [3]Department of Medical Sciences, Molecular Epidemiology, Uppsala University, Uppsala, Sweden. [4]Division of Cardiology, Department of Medicine, Karolinska Institutet, Stockholm, Sweden. [5]School of Public Health and Community Medicine, Institute of Medicine, University of Gothenburg, Gothenburg, Sweden. [6]Department of Clinical Sciences, Danderyd University Hospital, Karolinska Institutet, Stockholm, Sweden. [7]Department of Public Medicine and Clinical Health, Umeå University, Umeå, Sweden. [8]Region Västra Götaland, Sahlgrenska University Hospital, Department of Clinical Physiology, Gothenburg, Sweden. [9]Center for Medical Image Science and Visualization (CMIV), Linköping University, Linköping, Sweden. [10]Department of Clinical Physiology in Linköping, Linköping University,

Linköping, Sweden. [11]Department of Health, Medicine and Caring Sciences, Linköping University, Linköping, Sweden. [12]Department of Clinical Sciences Lund, Cardiology, Lund University, Lund, Sweden. [13]Department of Medical Sciences, Clinical Epidemiology, Uppsala University, Uppsala, Sweden. [14]Department of Cardiology, Karolinska University Hospital, Stockholm, Sweden. [15]Department of Clinical Sciences in Malmö, Lund University, Malmö, Sweden. [16]Department of Emergency and Internal Medicine, Skåne University Hospital, Malmö, Sweden. [17]Region Västra Götaland, Sahlgrenska University Hospital, Department of Cardiology, Gothenburg, Sweden. [18]Department of Medical Sciences, Uppsala University, Uppsala, Sweden. [19]The George Institute for Global Health, University of New South Wales, Sydney, Australia. [20]Department of Biomedical and Clinical Sciences, Division of Clinical Genetics, Linköping University, Linköping, Sweden. ✉e-mail: anders.gummesson@vgregion.se

