## [Transparent Peer Review file · Nature Communications]

A genome-wide association study of imaging-defined atherosclerosis

Corresponding Author: Dr Anders Gummesson

Version 0:

Reviewer comments:

Reviewer #2

(Remarks to the Author)

Based on the Swedish CARDioPulmonary bioImage Study (SCAPIS), a prospective, large-scale population-based cohort mainly of European ancestry, the authors carried out a genome-wide association study (GWAS) of imaging-defined atherosclerosis traits in the coronary and carotid arteries, including 1) directly measured coronary plaque burden, defined as segment involvement score (SIS), by coronary computed tomography angiography (CCTA), 2) coronary artery calcification, defined as coronary artery calcium score (CACs), and 3) carotid plaque (CarPlaq) assessed by ultrasound. They identified 20 significant independent genetic markers for SIS, three of which were not implicated in previous genetic studies of atherosclerotic cardiovascular disease (ASCVD). They found highly similar genetic underpinnings between the two coronary traits, while both shared and non-shared loci were identified between SIS and CarPlaq. They also found that the genetic correlations of image-defined atherosclerosis variables with lipid-related risk factors were not significant, and thus the genetic factors can capture additional risk information beyond traditional risk factors for ASCVD. Finally, they quantify the effect of coronary plaque burden on risk of myocardial infarction (MI) by two-sample Mendelian randomization. The manuscript is well organized and written and could serve as an in-time resource for the cardiovascular disease research community. We only have a few comments and questions, which we hope may help further improve the manuscript.

1. I am missing what portion of the 27,000(?) SCAPIS individuals had measurable CAD (i.e., segment involvement score (SIS), by coronary computed tomography angiography (CCTA), 2) coronary artery calcification, defined as coronary artery calcium score (CACs), and 3) carotid plaque (CarPlaq). From Extended Table 1 it seems only 1/3 had "Any atherosclerosis – what is the definition of any and what stage does that correspond to?"

What portion of the patients had clinically significant CAD? 0%?

As SCAPIS recruit healthy individuals and the measurements reported were assessed at baseline _ I am guessing the measured CAD was largely very mild and far from clinically significant? So additional information about the clinical status of the SCAPIS individuals at the time of the angiographic assessments are needed. Please also comment what likely relevance GWAS loci for very early and mild non-clinical CAD may be (if any) for later risk of developing manifest clinically significant CAD? Could it be that this GWAS study of SCAPIS is premature?

2. Have some or all of these 27,000 SCAPIS individuals been at the first 5-year follow-up? If yes – does the reported loci remain significant? Even if only a smaller fraction has been evaluated at the 5-year follow up – it would serve as a strong validation.

3. In GWAS analyses section of Methods: why did age need to be modelled with restricted cubic splines here? Can you provide more method details or data/plots to help audience understand this?

4. In Fig 1A: We would suggest putting SIS and CACS GWAS results in different panels of Manhattan plots rather than clump them together to make them more legible.

5. In Table 1: It may be informative to add to each of the previously implicated SNPs the associated ASCVD traits, effect size and direction, p-value, and literature references, so that it is readily for the audience to refer to existing knowledge and compare the GWAS results from the current study with previous studies.

6. In Assessment of potential effect mediators section of Methods and Results:

- 1) Why was the mediation analysis performed by adjusting the potential risk factors in regression analysis rather than using a designated mediation analysis method, e.g., *bmediatR* (Crouse, et al. *PLoS Genet.* 2022; PMID: 35533209)?
- 2) Why were the 6 traditional risk factors chosen in the mediation analysis shown in Fig 3A among others included in Fig 3B? For example, hypertension is genetic correlation with all the 3 image-defined ASCVD traits (Fig 3B), but why was it not considered as a potential mediator in Fig 3A?
- 3) The genetic correlation between Lp(a) and LDL-C and the 3 ASCVD traits are not significant and not as high as BMI (Fig 3B), but why did they appear more likely to potential mediators here? How to interpret this result?

7. In Fig 3B: Why is the r_g value of genetic correlation between heredity of MI and the 3 ASCVD variables are much higher (~0.6 - 0.8) than other risk factors, but not as significant? How to interpret this result?

8. The novel GWAS loci should be replicated using independent samples.

9. The LD score regression analysis (genetic r^2) was limited to variables and data points measured in SCAPIS. This analysis should be extended to large public GWAS summary-level data (e.g. *CardiogramplusC4D*).

10. I wonder if the angiographic traits measured in SCAPIS were compared to other angiographic traits, e.g. SYNTAX, in terms of phenotypic correlation and LDSC r^2 . Furthermore, the GWAS signal reported in this paper could be integrated with angiographic trait molecular signature (e.g. gene expression signal in relevant tissue) to obtain insights of pathogenesis. Carotid stenosis studies like *AtheroExpress* and *CVpath* may also be useful to better understand underlying mechanisms. The authors used GTEx for relation to RNA expression but GTEx does not have relevant clinical data as the biopsies were apprehended post mortem. Other relevant functional studies to validated atherosclerosis-related SNPs is for example the *STARNET* study with detailed CAD measurements.

Reviewer #3

(Remarks to the Author)

In their study, Anders Gummesson et al. performed three GWAS in a large population-based cohort using CCTA data. The phenotypes are i) segment involvement score, ii) coronary artery calcification, and iii) ultrasound-assessed carotid plaques. They identified 20 independent loci for SIS, three of which have not been associated with ASCVD. Finally, they performed two-sample MR analyses of SIS-associated and MI-associated variants using two prior GWAS of MI. They estimated that a 1-unit increase in SIS corresponded to 1.8-fold increased odds of MI.

Overall, the manuscript is well-written, and clear to read. The GWAS are properly conducted and the authors show good attention to detail in their analyses. The GWAS summary statistics will be invaluable for downstream analyses if made publicly available. However, I am concerned about the validity and interpretation of the MR analyses.

Major comments:

Although there is a convincing visual relationship between SIS and MI in Figure 4, the authors could provide more details on the MR analyses, perform sensitivity analyses (for example, MR-Egger, leave-one-out), and justify that all 3 MR assumptions have been met (especially #2).

Given that the MR estimate is high, at least compared to previous meta-analyses (as the authors point out), I suggest that the authors discuss the validity of their MR analyses, especially if they satisfy "MR assumption #2". Are the IVs "associated with the outcome only through the exposure" (e.g., see Bell et al., <https://doi.org/10.1161/CIRCOUTCOMES.119.005623>)? In particular, the SNP rs186696265, rs10455872, and rs7412 have been associated with traits such as Lp(a) and LDL-C before (e.g., GWAS Catalog).

The authors could give the MR-Egger estimate in addition to the IVW estimate and test for horizontal pleiotropy using the MR-Egger intercept. If some SNPs directly affect MI, that is, if their effect on MI is not 100% mediated through SIS, then the MR estimate may be upward biased. In other words, the authors should justify why they think that the direct effect "SNP → mediator → MI", where "mediator" is Lp(a), LDL-C, etc., is low / not problematic in the case of rs186696265, rs10455872, and rs7412.

Also: The effect sizes for SIS in Figure 4 do not match the effect sizes reported in Table 1. For example, rs186696265 has Beta=0.5 in Table 1 but ≈0.4 in Figure 4. In the legend for Figure 4, the authors state that SIS is "quantified as a continuous variable" but do not explain this in the Methods section.

Minor comments:

The authors could highlight the usefulness of their GWAS summary statistics for PRS development. Most current PRS for CAD are constructed from hard events, so they often include non-specific variants, that is, variants associated with risk factors (e.g., blood pressure, cholesterol). As GWAS of SIS, CACS, and CarPlaq are more specific, they could be valuable for more specific PRS for CAD.

p13, l273: "Annotation of loci were based on the protein-coding gene with the shortest distance to the 274 lead SNP.". The authors could mention that they also annotated variants with Open Targets and GTEx to annotate the genes (provided in

Extended Data Table 2).

p14, l280: "r package" → "R package"

p18, l379: Correlated significantly: How was significance defined?

p31: Regarding Figure 4: Is there a specific reason why these SNPs are annotated?

p31, l637: "Quantified as a continuous variable"? What does this mean? Wasn't this variable treated as an ordinal variable in the GWAS?

Reviewer #4

(Remarks to the Author)

Version 1:

Reviewer comments:

Reviewer #2

(Remarks to the Author)

The authors addressed most of my questions and presented a revised manuscript. However, I still have concerns related to my comment#1. This paper identifies significant GWAS signals on three imaging traits: SIS, CACS and CarPlag (shown in Table 1), where the SIS analysis yielded most GWAS peaks. More importantly, three SIS peaks were not previously identified by ASCAD GWAS.

Literature reported CACS as a predictor of ASCAD risk, which is mentioned by the authors and referenced Detrano et al. (N Engl J Med. 2008. PMID: 18367736). In contrast, SIS has not been convincingly linked to ASCAD risk. In clinical practice, SIS is not an important predictor for ASCAD.

Reviewer #3

(Remarks to the Author)

The authors have responded well to the suggestions.

Reviewer #4

(Remarks to the Author)

Below is our point-by-point response to Reviewers. We thank the reviewers for thorough scrutiny of our work, which we believe has helped improve the manuscript considerably. We should also comment that in addition to the changes in the manuscript mentioned below, there have also been changes not mentioned below that were due to journal policies and guidelines.

REVIEWER COMMENTS

Reviewer #1 (Remarks to the Author):

Based on the Swedish CARDioPulmonary bioImage Study (SCAPIS), a prospective, large-scale population-based cohort mainly of European ancestry, the authors carried out a genome-wide association study (GWAS) of imaging-defined atherosclerosis traits in the coronary and carotid arteries, including 1) directly measured coronary plaque burden, defined as segment involvement score (SIS), by coronary computed tomography angiography (CCTA), 2) coronary artery calcification, defined as coronary artery calcium score (CACs), and 3) carotid plaque (CarPlaq) assessed by ultrasound. They identified 20 significant independent genetic markers for SIS, three of which were not implicated in previous genetic studies of atherosclerotic cardiovascular disease (ASCVD). They found highly similar genetic underpinnings between the two coronary traits, while both shared and non-shared loci were identified between SIS and CarPlaq. They also found that the genetic correlations of image-defined atherosclerosis variables with lipid-related risk factors were not significant, and thus the genetic factors can capture additional risk information beyond traditional risk factors for ASCVD. Finally, they quantify the effect of coronary plaque burden on risk of myocardial infarction (MI) by two-sample Mendelian randomization. The manuscript is well organized and written and could serve as an in-time resource for the cardiovascular disease research community. We only have a few comments and questions, which we hope may help further improve the manuscript.

1. I am missing what portion of the 27,000(?) SCAPIS individuals had measurable CAD (i.e., segment involvement score (SIS), by coronary computed tomography angiography (CCTA), 2) coronary artery calcification, defined as coronary artery calcium score (CACs), and 3) carotid plaque (CarPlaq).

From Extended Table 1 it seems only 1/3 had “Any atherosclerosis – what is the definition of any and what stage does that correspond to?”

What portion of the patients had clinically significant CAD? 0%?

As SCAPIS recruit healthy individuals and the measurements reported were assessed at baseline _ I am guessing the measured CAD was largely very mild and far from clinically significant? So additional information about the clinical status of the SCAPIS individuals at the time of the angiographic assessments are needed. Please also comment what likely relevance GWAS loci for very early and mild non-clinical CAD may be (if any) for later risk of developing manifest clinically significant CAD? Could it be that this GWAS study of SCAPIS is premature?

Thank you for these valuable comments. We agree that this should be better clarified in the manuscript, and in the Results/Study population we have therefore added a sentence: “Of these, 10,589 (42.7%) had SIS>0, 10,645 (40.9%) had CACS>0 and 14,813 (55.3%) had CarPlaq>0, respectively (**Extended Table 1**).” In Extended table 1, we have added more information to clarify that “any” is defined as defined as SIS>0, CACS>0 and CarPlaq>0,

respectively. In the same table, we have also added more granular information on the distribution of SIS, CACS and CarPlaq, as well as information on the proportion of subjects with obstructive coronary atherosclerosis (at least one vessel with >50% stenosis).

Being a population-based cohort, SCAPIS includes a range of CAD severity. Also mild to moderate atherosclerosis can contribute information in the analysis since it confers a significantly increased risk of CVD outcomes. For example, a study in MESA reported an increased risk of myocardial infarction or CAD-related mortality by a factor of 3.89 (95% CI 1.72-8.79, $P < 0.001$) in the lowest range of positive CACS (i.e., 1-100) compared to participants with CACS=0 (Detrano et al. N Engl J Med. 2008. PMID: 18367736). Mild atherosclerosis is a relevant outcome in genetic association studies as they can disclose novel mechanisms involved in the earliest phase of atherosclerosis development. It is therefore relevant to include all atherosclerosis measurements in the analysis, while at the same time taking the range into account by using ordinal or linear regression rather than logistic regression.

To comment on this, we have added the following sentence in the discussion: “Although detectable coronary plaques were common (42.7%), most were of mild to moderate severity and relatively few had obstructive CAD (5.7%). Nevertheless, also mild atherosclerosis is of relevance as it confers a considerably increased risk of CVD events (Detrano et al. N Engl J Med. 2008. PMID: 18367736). Most importantly, genetic association data of mild atherosclerosis may help to elucidate mechanisms involved in the early phases of atherosclerosis development.”

We do not think that this GWAS study of SCAPIS is premature, for several reasons. In the 50-64 age range, at least half of the population has signs of atherosclerosis while ASCVD events is still relatively rare, hence this age range was deemed optimal for the overall objectives of SCAPIS to prospectively study subclinical ASCVD in relation to future events. Choosing an older population would certainly have increased the number of study participants with clinically overt ASCVD, but had been less relevant for the purpose of early detection. In addition, there are potential issues with older populations. Firstly, the relative importance of genetics is likely to be higher in younger age and diluted by life-style factors as age increases. Secondly, older age-groups are more likely to be depleted of severe cases – e.g. when patients have died before being enrolled (survivor bias).

Taken together, we believe that for the purpose of studying the genetics of subclinical atherosclerosis, the 50-64 age range is likely optimal. The future might bring additional studies in other cohorts of older ages that can complement our findings.

2. Have some or all of these 27,000 SCAPIS individuals been at the first 5-year follow-up? If yes – does the reported loci remain significant? Even if only a smaller fraction has been evaluated at the 5-year follow up – it would serve as a strong validation.

The baseline is the only timepoint where CCTA and carotid ultrasound will be available for the whole cohort. Although there will be a 10-year follow up examination that includes CCTA, it will only include half ($n \sim 15,000$) of the study population, hence it will be less suitable for a GWAS. These examinations started earlier this year and it will be another four years of study visits followed by time for data curation, so at least five years before we can use the data.

The next set of data to be added in SCAPIS is register data on clinical events, but that is not yet available. On the other hand, the main scope of our study is imaging-defined atherosclerosis and not events. Furthermore, a GWAS on events in SCAPIS would be

dwarfed by the data that we already use in the article (i.e. UK Biobank and CARDIoGRAMplusC4D).

It is possible that the discussion-point in the previous version, i.e. "*A more direct comparison will be possible in a few years when prospective outcome data is available in SCAPIS*", might have caused confusion - this sentence referred to future observational studies in SCAPIS, which was not so relevant and hence removed.

3. In GWAS analyses section of Methods: why did age need be modelled with restricted cubic splines here? Can you provide more method details or data/plots to help audience understand this?

The assumption that age has a linear association with the outcome is strong, and probably untenable except as a rough approximation. Age can also be categorized, but using age categories lose information and power, and may increase the risk of residual confounding. We could also have adjusted for age using a square term, but such adjustments are in general more volatile and influenced by single outliers compared to cubic splines adjustment. Using splines does have the drawback of increased model complexity, but a restricted cubic spline with four knots means three variables for age, compared to two variables (using a square term) or one variable (linear). Therefore, the spline adjustment doesn't contribute much to the model complexity compared to, for example, the 10 additional genetic principal components. Also, since our sample size is large and our outcomes are fairly common we don't expect the models to become unstable due to the extra number of variables.

We have added the following sentence in the manuscript to provide a rational for cubic splines: " Age was modelled with restricted cubic splines (using four knots at the 5th, 35th, 65th and 95th percentile) to relax the assumption of linearity and thereby potentially improve power (Kahan et al. BMC Med Res Methodol. 2016 Apr 11:16:42. PMID: 27068456)".

4. In Fig 1A: We would suggest putting SIS and CACS GWAS results in different panels of Manhattan plots rather than clump them together to make them more legible.

We agree with this suggestion and have created a new figure with the Manhattan plots separated (see below). This opened up a white space, so we moved the QQ-plot (previously provided as an extended figure) to fill that space.

5. In Table 1: It may be informative to add to each of the previously implicated SNPs the associated ASCVD traits, effect size and direction, p-value, and literature references, so that it is readily for the audience to refer to existing knowledge and compare the GWAS results from the current study with previous studies.

We agree that this would be informative. The large number of studies conducted in this area makes it challenging to summarize all previous knowledge in a brief way. Many of the SNPs have been linked to ASCVD traits in several studies, so providing effect sizes and p-values from all of these studies is not feasible. Also, not all SNPs have not been directly reported to associate with ASCVD in the GWAS Catalog, but are instead linked via one or several proxies, i.e. other SNPs in the same locus.

Given these challenges, we decided to extract data from three recent large GWAS studies: Aragam et al (CAD events), Kavousi et al (CACS), and Yeung 2021 (carotid IMT). Even when limiting to such few studies, the table becomes extensive and is therefore provided as supplementary data. This data is also useful for lookups (relating to comment #8 about the need for validation, see below), and also provides data on cIMT which was lacking in our

study. Furthermore, this gave us the opportunity to provide more complete data from our study, i.e. all betas and p-values for all outcomes, in a format easier for other researchers to extract as compared to the Table 1 in the manuscript.

We have referenced the table the following way under “GWAS Results”: “For all SNPs associated with any outcome, data on associations with SIS, CACS and CarPlaq, together with data from three recent GWAS studies on CACS, CAD events and cIMT, are provided as supplementary data.”

For the reader who wants more information, it is readily available in public databases which are updated regularly, as opposed to our article.

6. In Assessment of potential effect mediators section of Methods and Results: 1) Why was the mediation analysis performed by adjusting the potential risk factors in regression analysis rather than using a designated mediation analysis method, e.g., bmediatR (Crouse, et al. PLoS Genet. 2022; PMID: 35533209)?

Our reasoning was that, if we could, we should use a type of analysis that is easy to understand for the reader. Most readers are familiar with regression analysis and adjusting for co-variates. However, the reviewer is correct to point out that there are alternative methods. Therefore, we performed mediation analysis using bmediatR upon reviewer request. The results are broadly concordant using either method, which strengthens the overall results. Briefly, for 18 of 20 SNPs, both methods agree: the effect on coronary atherosclerosis for rs10455872 and rs1866962652 are mediated by Lp(a) (see figure below, A and B), and for 16 SNPs there is no mediatory effect of any of the risk factors (data not shown).

Lastly, for 2 SNPs the results are less clear: For rs7412 (APOE), regression results indicate a partial mediation of LDL-C, and such a tendency is also seen for rs28601761 (TRIB1), whereas bmediatR suggests mediation for rs28601761 via TG (see figure below, C and D). Hence, the combined results for these 2 SNPs are not as conclusive as for the other SNPs and should therefore be interpreted with caution.

Taken together results are conclusive for all SNPs but rs7412 and rs28601761, and we used the following wording to reflect this:

- Results: “Adjustment with Lp(a) attenuated the effect sizes for the two SNPs located in the LPA-PLG locus (rs10455872 and rs186696265). No other clear effect mediators were observed apart from a tendency towards lipid-mediation for 28601761 and rs7412.”
- Discussion: “The only clear mediating effect of established risk factors in our data was circulating Lp(a) levels for the LPA-PLG locus.”
- Also, a sentence about this in the legend were removed since Nat Com author guidelines not to include comment about results in figure legends.

Below, we include the requested bmediatR-based analyses with regards to the two Lp(a)-associated SNPs and the two SNPs in APOE and TRIB1 for the reviewer to examine. For sake of brevity, we opted to not include these analyses in the manuscript, but we thank the reviewer for this constructive suggestion.

2) Why were the 6 traditional risk factors chosen in the mediation analysis shown in Fig 3A among others included in Fig 3B? For example, hypertension is genetic correlation with all the 3 image-defined ASCVD traits (Fig 3B), but why was it not considered as a potential mediator in Fig 3A?

In Fig 3A, more variables would make the graph very crowded and difficult to read. We reasoned that the chosen variables HbA1c, BMI, and SBP was sufficient to account for the risk factors of diabetes, obesity and hypertension, respectively. None of these variables were found to be mediators for any SNP and that conclusion remains the same for the closely related traits waist circumference, fasting glucose, diabetes diagnosis, and hypertension diagnosis. As the reviewer points out, hypertension diagnosis had relatively high genetic correlations with all ASCVD traits; However, this is also true for SBP which has highly similar r_g -values. Hence, for the sake of readability, we suggest keeping it as is.

3) The genetic correlation between Lp(a) and LDL-C and the 3 ASCVD traits are not significant and not as high as BMI (Fig 3B), but why did they appear more likely to potential mediators here? How to interpret this result?

The mediation analysis in Fig 4A focuses on one SNP at a time and tests for potential mediating effects of each risk factor for that specific SNP. In contrast, the genetic correlations in Fig 3B are estimated based on the genome-wide genotype data and estimate the how much overall heritability is shared between two traits. Hence, the results of these vastly different analyses cannot be compared.

We acknowledge that showing these two figures side by side might create confusion, and we have therefore separated them into separate figures: The mediation analysis is now Fig.3, and the genetic correlation analysis is Fig 4. In addition, we have modified the figure legends to point out that mediation analysis was performed on specific SNPs and that genetic correlations are genome-wide.

Please note that the genetic correlations have been extended with external GWAS data in accordance with comment #9 below, hence it now has two panels.

7. In Fig 3B: Why is the rg value of genetic correlation between heredity of MI and the 3 ASCVD variables are much higher (~0.6 - 0.8) than other risk factors, but not as significant? How to interpret this result?

While the high point estimate might give the impression of being significant, the confidence interval is very large, as shown in the figure enclosed in the response to comment #9. This is due to the relatively few individuals with heredity for MI in the whole SCAPIS cohort (n=1822, 6.9%) for GWAS, as reported in Extended Table 1.

8. The novel GWAS loci should be replicated using independent samples.

We do agree with the Reviewer on this point. Validation of the novel associations with SIS would require a decently sized population with CCTA-data, which unfortunately is not available. The second largest cohort with CCTA data is the Copenhagen General Population Study (n=9533) (Fuchs et al, Ann Intern Med. 2023 Apr;176(4):433-442. PMID: 36972540), but we are not aware of them having genetics data ready to share. Hence, the main strength of this study, i.e the unique CCTA data, is also its main weakness. Being aware of the importance of validation, we chose not to emphasize these novel SNPs too much, and also mentioned it as a limitation in the discussion. We believe that the main importance of our study is to complement existing GWAS data on events with that of directly measured atherosclerosis, rather than identifying novel ASCVD loci.

In the absence of SIS GWAS data from independent cohorts, the second-best option for external validation could be CACS, while bearing in mind that theoretically there could still be SNPs that affect SIS but not CACS. Since the study by Kavousi et al is the so far largest CACS GWAS, we decided to perform look-ups in their data (now provided as supplementary data, see also comment #5). However, we refrain from using the word validation in the manuscript since it is not the same outcome variable.

The strongest support was for rs12190315, and we therefore expanded on this in the discussion: "In a look-up of these SNPs using CACS GWAS data by Kavousi et al, only rs12190315 had a low (albeit not genome-wide significant) p-value of 6.3×10^{-6} "

(Supplementary data). Hence, additional studies in independent cohorts with CCTA-data would be needed to validate our findings.”

Since the lookup for the other two novel SNPs gave none (rs17127656) or limited (rs2594811) support, we kept the discussion around them very brief as before, just emphasizing the need for validation.

9. The LD score regression analysis (genetic r^2) was limited to variables and data points measured in SCAPIS. This analysis should be extended to large public GWAS summary-level data (e.g. CardioGramPlusC4D).

We thank the Reviewer for the very valuable suggestion. While analyses of genetic correlations within SCAPIS have some advantages such as a large set of variables and no bias due to differences in study design, it has the disadvantage of being relatively small as compared to large public GWAS data. Doing both can therefore provide a more complete picture.

We have updated the genetic correlation analysis using external large consortium GWAS summary statistics, including:

- CardioGramPlusC4D
- Global Lipid Genetic Consortium (GLGC)
- International Consortium for Blood Pressure (ICBP)
- Meta-Analyses of Glucose and Insulin-related traits Consortium (MAGIC)
- The Genetic Investigation of Anthropometric Traits (GIANT)

In the revised version of the manuscript, we present genetic correlations in Fig 4, including based on SCAPIS in panel A and external data in panel B, see below.

Legend: "Figure 4. Genome-wide genetic correlation (rg) between atherosclerosis variables and other ASCVD-related phenotypes. A) Genetic correlations with phenotypes within SCAPIS. B) Genetic correlations using GWAS data from external consortia. Numbers in bold refer to genetic correlations with P-value < 0.05."

In the Results section, the wording now reads: "Genetic correlations across atherosclerosis variables and ASCVD-related traits using GWAS data generated within SCAPIS (Fig.4A) and data from external GWAS consortia (Fig.4B). The genetic correlations between atherosclerosis traits (SIS, CACS, CarPlaq and CAD events) were generally high, whereas for ASCVD risk factors such as BMI, SBP, HbA1c and lipids, the genetic correlations with atherosclerosis variables were relatively modest overall."

For review purposes, we also provide the figure below showing also the confidence intervals (please also see answer to point 3). As seen here, the correlation coefficients are largely comparable between SCAPIS and the data obtained from the consortia, but the confidence interval are considerably larger in SCAPIS because of the smaller sample size. We can also see that the genetic correlations with LDL-C are likely to be stronger than suggested by SCAPIS data only. This may be due to the differences in study size but also study design.

10. I wonder if the angiographic traits measured in SCAPIS were compared to other angiographic traits, e.g. SYNTAX, in terms of phenotypic correlation and LDSC r^2 . Furthermore, the GWAS signal reported in this paper could be integrated with angiographic trait molecular signature (e.g. gene expression signal in relevant tissue) to obtain insights of pathogenesis. Carotid stenosis studies like AtheroExpress and CVpath may also be useful to better understand underlying mechanisms. The authors used GTEx for relation to RNA expression but GTEx does not have relevant clinical data as the biopsies were apprehended post mortem. Other relevant functional

studies to validated atherosclerosis-related SNPs is for example the STARNET study with detailed CAD measurements.

We thank the Reviewer for these very relevant observations. The CCTA data recorded in our study only include presence of an atherosclerotic plaque (>/<50%). This limits our possibility to derive previously published angiographic risk scores based on invasive coronary angiography such as SYNTAX. Further, to the best of our knowledge SYNTAX has not been validated for grading of coronary lesions with CCTA (Kerner A et al. EuroIntervention 2013; 8:1428-34, PMID 23680958).

The point on insights on pathogenesis is very well taken and we agree with the Reviewer that identification and validation of eQTLs in multiple datasets reinforce our data. To comply with the Reviewer concern, we have added the information on eQTLs in atherosclerotic aortic wall in STARNET (n=600, Franzén et al. Science. 2016 Aug 19;353(6301):827-30. PMID: 27540175) in Extended Data Table 3. This set of data resemble most the stage of disease we study in SCAPIS.

Reviewer #2 (Remarks to the Author):

In their study, Anders Gummesson et al. performed three GWAS in a large population-based cohort using CCTA data. The phenotypes are i) segment involvement score, ii) coronary artery calcification, and iii) ultrasound-assessed carotid plaques. They identified 20 independent loci for SIS, three of which have not been associated with ASCVD. Finally, they performed two-sample MR analyses of SIS-associated and MI-associated variants using two prior GWAS of MI. They estimated that a 1-unit increase in SIS corresponded to 1.8-fold increased odds of MI.

Overall, the manuscript is well-written, and clear to read. The GWAS are properly conducted and the authors show good attention to detail in their analyses. The GWAS summary statistics will be invaluable for downstream analyses if made publicly available. However, I am concerned about the validity and interpretation of the MR analyses.

Major comments:

Although there is a convincing visual relationship between SIS and MI in Figure 4, the authors could provide more details on the MR analyses, perform sensitivity analyses (for example, MR-Egger, leave-one-out), and justify that all 3 MR assumptions have been met (especially #2).

Given that the MR estimate is high, at least compared to previous meta-analyses (as the authors point out), I suggest that the authors discuss the validity of their MR analyses, especially if they satisfy "MR assumption #2". Are the IVs "associated with the outcome only through the exposure" (e.g., see Bell et al., <https://doi.org/10.1161/CIRCOUTCOMES.119.005623>)? In particular, the SNP rs186696265, rs10455872, and rs7412 have been associated with traits such as Lp(a) and LDL-C before (e.g., GWAS Catalog).

The authors could give the MR-Egger estimate in addition to the IVW estimate and test for horizontal pleiotropy using the MR-Egger intercept. If some SNPs directly affect MI, that is, if their effect on MI is not 100% mediated through SIS, then the MR estimate may be upward biased. In other words, the authors should justify why they

think that the direct effect “SNP → mediator → MI”, where “mediator” is Lp(a), LDL-C, etc., is low / not problematic in the case of rs186696265, rs10455872, and rs7412.

We thank the reviewer for these observations. We agree that the MR analyses are subject to potential bias, as many MR analyses are. The basic hypothesis we test is whether coronary plaques are causally related with myocardial infarction. Arguably, this hypothesis is very plausible, and it is more a question of the magnitude of the effect rather than the direction. We do agree that there is potential for violation of assumption #2, more specifically, it is conceivable that some SNPs may affect risk of MI by other means than only by affecting coronary plaque development. If so, may be revealed by the MR Egger analysis as an OR above 1 (equivalent to a positive intercept when expressed as the logOR) if these effects are systematic across SNPs.

To test this, we conducted MR Egger analysis, alongside several other methods, shown below. We did see a significant intercept using the MR-Egger method (P=0.015), however the OR intercept itself was below 1, indicating a slight protective effect which is in the opposite direction of the reasoning above. Furthermore, the MR-Egger estimate was higher than using IVW (2.52 vs 1.83). Hence, these analyses suggest that the OR from IVW may be a conservative estimate rather than being upward biased.

As a reference, we also included several other MR methods and the odds ratios were all in a similar range as the main analysis, although the MR-Egger methods tended to be higher. We have now included the table with the different MR methods as Extended data table 3 and added the following text in the main manuscript: *“Sensitivity analysis in UK Biobank using the MR-Egger method, to account for potential pleiotropic effects, resulted in a higher estimate (OR 2.52, 95% CI: 1.89-3.37), hence it is possible that the IVW may slightly underestimate the true SIS-associated MI risk. For further details on sensitivity MR analyses, see Extended Table 3”*.

It is also commented in the discussion: *“Of note, our sensitivity analyses with MR-Egger indicate that the OR from the main analysis may be a conservative estimate rather than being upward biased.”*

Number of SNPs = 20, MR analysis using SCAPIS and UKBiobank				
Method	Odds Ratio	95% CI		P-value
Simple median	1.77	1.59	1.97	4.84E-25
Weighted median	1.85	1.69	2.03	5.16E-39
Penalized weighted median	1.85	1.69	2.04	8.02E-38
IVW	1.83	1.59	2.1	2.65E-17
Penalized IVW	1.82	1.73	1.92	6.52E-108
Robust IVW	1.8	1.57	2.06	5.26E-17
Penalized robust IVW	1.84	1.75	1.94	1.84E-116
MR-Egger	2.52	1.89	3.37	3.00E-10
(intercept)	0.952	0.916	0.99	0.015
Penalized MR-Egger	2.29	1.8	2.92	1.93E-11
(intercept)	0.963	0.936	0.991	0.010
Robust MR-Egger	2.44	1.7	3.52	1.53E-06
(intercept)	0.956	0.914	1	0.048
Penalized robust MR-Egger	2.23	1.78	2.8	3.60E-12

(intercept)	0.967	0.936	1	0.049
-------------	-------	-------	---	-------

To address the Reviewer's concern about the SNPs rs186696265, rs10455872, and rs7412, we performed the analysis without these SNPs (see table below, not included in the manuscript). In this analysis there was no longer a significant intercept. The estimate was only marginally lower. (OR=1.66, 95% CI: 1.47-1.88) and the MR-Egger estimate was higher (OR=2.01, 95% CI: 1.46-2.76) as compared to the main IVW analysis with all 20 SNPs.

Number of SNPs = 17 (excluding rs7412, rs186696265 and rs10455872), MR analysis using SCAPIS and UKBiobank				
Method	Odds Ratio	95% CI		P-value
Simple median	1.72	1.51	1.96	1.69E-16
Weighted median	1.81	1.65	1.99	6.74E-34
Penalized weighted median	1.83	1.66	2.02	1.32E-34
IVW	1.66	1.47	1.88	1.49E-15
Penalized IVW	1.8	1.69	1.91	1.87E-80
Robust IVW	1.67	1.44	1.94	2.81E-11
Penalized robust IVW	1.81	1.71	1.92	1.99E-91
MR-Egger	2.01	1.46	2.76	1.75E-05
(intercept)	0.975	0.937	1.01	0.205
Penalized MR-Egger	1.92	1.63	2.26	2.44E-15
(intercept)	0.982	0.962	1	0.085
Robust MR-Egger	1.99	1.69	2.35	2.22E-16
(intercept)	0.976	0.949	1	0.089
Penalized robust MR-Egger	1.93	1.72	2.16	<1E-100
(intercept)	0.984	0.958	1.01	0.227

To further expand on your points, we do think that the following causal chain is possible for a subset of the SNPs: SNP → mediator → SIS → MI, where mediator is e.g. LDL-C or Lp(a). The plots below (not included in the manuscript) show all independent significant SNPs for LDL-C (left panel) or Lp(a) (right panel) in UK Biobank. Rs7412 mainly associates with LDL-C, whereas rs186696265 and rs10455872 associates mainly with Lp(a). The SNPs are plotted based on their association with SIS and MI, respectively. The results support the SNP → LDL/Lp(a) → SIS → MI pathway, and therefore, violation of assumption #2 seems unlikely.

Taken together, our assessment is that the pathways SNP → risk factor → SIS → MI and/or SNP → SIS → MI are the most plausible causal pathways to explain the MR model results. We cannot completely rule out the existence of any SNPs affecting both SIS *and* another factor that directly affects MI (i.e. affects MI without affecting SIS), but evidently this risk is low, and it is unlikely to bias the overall MR-estimate (based on all 20 SNPs) very much.

Also: The effect sizes for SIS in Figure 4 do not match the effect sizes reported in Table 1. For example, rs186696265 has Beta=0.5 in Table 1 but ≈0.4 in Figure 4. In the legend for Figure 4, the authors state that SIS is “quantified as a continuous variable” but do not explain this in the Methods section.

We apologize for being unclear. In order to perform the MR-analysis and make it interpretable, the exposure-variable is preferably expressed as a continuous variable. For example, if the exposure would be LDL-C or blood pressure, you want to express the units in mmol/L or mmHg because one wants to express the odds ratio for the outcome in terms of “per unit exposure”. SIS is an approximate continuous variable since it can vary between 0 and 11 and hence it is possible to express the risk of MI in terms of “per one extra segment”. For the main analysis (from which the SNPs are selected), SIS is coded as an ordinal variable hence the beta-coefficients will be different.

To clarify we have added the following sentences in the Methods: “Since there is already strong prior probability that coronary plaques are causally related to MI, the purpose of the MR analysis was primarily to quantify this relationship. Hence, to obtain an estimate expressed as the risk of MI per one extra segment with atherosclerosis, SIS was coded as a continuous variable in this analysis. The slope of the regression line between change in SIS and change in MI risk can be interpreted as the causal risk increase per one additional coronary segment with plaque.”

Minor comments:

The authors could highlight the usefulness of their GWAS summary statistics for PRS development. Most current PRS for CAD are constructed from hard events, so they often include non-specific variants, that is, variants associated with risk factors (e.g., blood pressure, cholesterol). As GWAS of SIS, CACS, and CarPlaq are more specific, they could be valuable for more specific PRS for CAD.

We fully agree that this GWAS data has a great potential for PRS development. For example, efforts are currently underway to apply PRS-triaged CACS measurements (Gray et al, Am Heart J. 2023 Oct;264:163-173. PMID: 37364748), and under such applications a more plaque-specific PRS could be evaluated. A properly conducted PRS study can be quite extensive and involve external validation cohorts, comparisons with benchmarks etc, so in our opinion it is better suited as an article of its own. We hope that we interpret the reviewer correctly that the comment is not a request to perform a PRS study within this article, but rather to highlight its potential for future studies.

We have added the following sentences in the discussion (right after the paragraph about the SIS vs MI relationship): “Another potential application for these GWAS data, outside the scope of the present study, is to use GWAS summary statistics to develop polygenic risk scores (PRS). Most current PRS for CAD are constructed from hard events, but for applications that aim to identify individuals with subclinical atherosclerosis, a more plaque-specific PRS could be evaluated.”

p13, I273: “Annotation of loci were based on the protein-coding gene with the shortest distance to the 274 lead SNP.”. The authors could mention that they also annotated variants with Open Targets and GTEx to annotate the genes (provided in Extended Data Table 2).

Thank you for this comment. As suggested by Reviewer #1, we have also added annotations from STARNET. In addition, we provide in the same sentence links to the data sources and references to main publications. The sentence now reads: “Gene annotations were also obtained from Open Targets (<https://www.opentargets.org>)²⁸, Genotype-Tissue Expression (GTEx, <https://www.gtexportal.org/home>)²⁹, and Stockholm–Tartu Atherosclerosis Reverse Network Engineering Task (STARNET, <http://starnet.mssm.edu>)³⁰ (Extended Table 2).”

p14, I280: "r package" → "R package"

Thank you for pointing out this misprint, it has now been corrected.

p18, I379: Correlated significantly: How was significance defined?

Significance was defined as $p < 0.05$. In the revised version, we specified this instead of using the word significant in this context. The figure legend now reads: “Numbers in bold refer to genetic correlations with P-value < 0.05.”

p31: Regarding Figure 4: Is there a specific reason why these SNPs are annotated?

This was a matter of aesthetics/readability – We thought that the figure became too crowded when all SNPs were annotated, and we therefore chose to only annotate the ones with the highest/lowest estimates. See below a version with all SNPs annotated, which we have not included in the revised version, although this is a possibility if preferred by the reviewer/editors.

p31, I637: "Quantified as a continuous variable"? What does this mean? Wasn't this variable treated as an ordinal variable in the GWAS?

Again, apologies for being unclear about this. Please see previous response above that aims to clarify.

Reviewer #3 (Remarks to the Author):

Thank you very much for valuable input! We think that this has helped improve the article considerably.

Response to the reviewers' comments

We thank the Reviewers for constructive feedback, which helped us to improve our work.

REVIEWERS' COMMENTS

Reviewer #2 (Remarks to the Author):

The authors addressed most of my questions and presented a revised manuscript. However, I still have concerns related to my comment#1. This paper identifies significant GWAS signals on three imaging traits: SIS, CACS and CarPlag (shown in Table 1), where the SIS analysis yielded most GWAS peaks. More importantly, three SIS peaks were not previously identified by ASCAD GWAS.

Literature reported CACS as a predictor of ASCAD risk, which is mentioned by the authors and referenced Detrano et al. (N Engl J Med. 2008. PMID: 18367736). In contrast, SIS has not been convincingly linked to ASCAD risk. In clinical practice, SIS is not an important predictor for ASCAD.

Reply: We thank the Reviewer for this comment. If we understand this correctly, the main concern relates to SIS as outcome variable its role as predictor of coronary events.

The main aim of this study is to elucidate the genetic basis of imaging-defined atherosclerosis. To this end, the CCTA in SCAPIS offers a direct measurement of coronary atherosclerosis that is unique for large cohorts. SIS is a simple and reliable tool that quantifies the number of coronary segments with detectable plaques, and there is a strong correlation between SIS and CACS (Bergström. et al. Circulation (2021) 144:916-929, reference included in the manuscript).

The lack of large cohorts with CCTA data has meant that much of the existing evidence around imaging-defined ASCAD in relation to ASCAD events is based on the more easily obtained CACS. Even so, the evidence linking SIS to ASCAD risk is not absent. In a systematic review and meta-analysis from 2017, involving 11 studies and 9777 subjects in total, it was concluded that SIS on coronary CTA is a strong, independent predictor of cardiovascular events (Ayoub et al. J Cardiovasc Comput Tomogr (2017) 11:258-267, reference included in the manuscript).

We agree that the role of SIS as predictor of coronary events needs to be further validated, and this is indeed a key rationale for our two-sample Mendelian randomization study that shows, and quantifies, a causal relationship between SIS and MI (Fig.5A,B). Please also note that the SIS-associated SNPs showed similar results for obstructive CAD (>50% stenosis) which is used for clinical decision making (Supp Fig.S2).

To comply with the Reviewer's concern, we have added a wording in the Discussion to acknowledge that the role of SIS as predictor of coronary events needs to be further validated: "*SIS is a direct measure of coronary plaques, and its role as predictor of coronary events needs to be further validated.*"

Reviewer #3 (Remarks to the Author):

The authors have responded well to the suggestions.

Reviewer #4 (Remarks to the Author):
